# Testing Assumptions Underlying a Unified Theory for the Origin of Grid Cells

**Rylan Schaeffer**[*]
Computer Science
Stanford University
rschaef@cs.stanford.edu

**Mikail Khona**[*]
Physics
MIT
mikail@mit.edu

**Adrian Bertagnoli**
Brain and Cognitive Sciences
MIT

**Sanmi Koyejo**
Computer Science
Stanford University
sanmi@cs.stanford.edu

**Ila Rani Fiete**
Brain and Cognitive Sciences
MIT
fiete@mit.edu

## Abstract

Representing and reasoning about physical space is fundamental to animal survival, and the mammalian lineage expresses a wealth of specialized neural representations that encode space. Grid cells, whose discovery earned a Nobel prize, are a striking example: a grid cell is a neuron that fires if and only if the animal is spatially located at the vertices of a regular triangular lattice that tiles all explored two-dimensional environments. Significant theoretical work has gone into understanding why mammals have learned these particular representations, and recent work has proposed a "unified theory for the computational and mechanistic origin of grid cells," claiming to answer why the mammalian lineage has learned grid cells. However, the Unified Theory makes a series of highly specific assumptions about the target readouts of grid cells - putatively place cells. In this work, we explicitly identify what these mathematical assumptions are, then test two of the critical assumptions using biological place cell data. At both the population and single-cell levels, we find evidence suggesting that neither of the assumptions are likely true in biological neural representations. These results call the Unified Theory into question, suggesting that biological grid cells likely have a different origin than those obtained in trained artificial neural networks.

## 1 Introduction

In the intricate realm of neural circuits that underpin navigation and spatial cognition, grid cells have emerged as an especially intriguing pattern of neuronal activity. Located in the mammalian medial entorhinal cortex, grid cells fire in a striking regular hexagonal grid pattern as an animal navigates through space [13]. Their unique firing properties, believed to represent a metric for spatial navigation, have drawn extensive attention. Recent work proposed a new "unified theory" for the origin of grid cells[2] [28, 30, 31, 29] to answer *why* the mammalian lineage has learned grid cells. However, the Unified Theory relies on a sequence of assumptions about grid cells performing supervised learning to predict specific targets, believed to be place cells (a type of neuron involved in spatial processing [24, 22, 23]). To our knowledge, these assumptions have not been tested in biological place cells. In this work, we seek to rectify this. We extract the assumptions made by the Unified Theory by

---

[*]Denotes equal contribution and co-first authorship.

[2]In this context, "theory" is intended in the sense of an accurate and predictive mathematical description of naturally occurring phenomena, akin to the theory of general relativity or quantum field theory.

NeurIPS 2021 AI for Science Workshop.

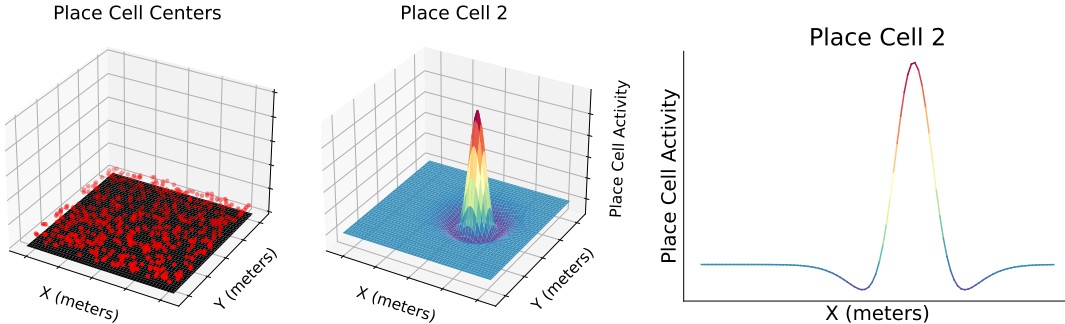

Figure 1: **Two key assumptions of the Unified Theory.** Left: Readouts, as a population, must be translationally invariant. Center and Right: Readouts, individually, must have center-surround tuning curves.

revisiting its derivation in detail (App. B) then hone in on two pivotal suppositions specifically related to the readouts i.e. supervised targets of biological grid cells. We evaluate these assumptions against data from biological place cells and find that both assumptions are likely false. Such conclusions challenge the Unified Theory's explanation for the origin of grid cells in mammals.

## 2    Results

**Identifying assumptions of the Unified Theory**    The Unified Theory seeks to answer why the mammalian lineage has learnt grid cells. Its answer is that grid cells are the optimal solution to predicting supervised targets that we generically call "readouts". Earlier papers claimed that these readouts biologically correspond to place cells [28, 30], although later papers [31, 29] suggested that these readouts might correspond to other biological quantities (more later). We reproduce the Unified Theory in detail (App. B) to highlight its assumptions. In this work, we focus on two:

1. The readouts, as a population, must be translationally invariant (Fig. 1, Left).
2. The readouts, individually, must have carefully tuned center-surround tuning curves: either Difference-of-Softmaxes (DoS) or a particular Difference-of-Gaussians (DoG) tuning curve shape (Fig. 1, Right); these functions are defined in App. A.

We focus on these two assumptions because they are mathematically critical for the Unified Theory's applicability to biological grid cells and numerically critical for deep recurrent neural networks to learn grid-like tuning [1, 28, 30, 21, 31], i.e., subsequent large-scale hyperparameter sweeps showed relaxing these assumptions causes disappearance of grid-like representations [26]. To the best of our knowledge, these assumptions have not been quantitatively tested in biological data; we do so here.

**Are place cells translationally invariant as a population?**    In order to explain the origin of grid cells, the Unified Theory requires that the readouts possess a translation-invariant spatial autocorrelation structure (App. B): only if the readouts' spatial autocorrelation is (approximately) Toeplitz will its eigenvectors be (approximately) Fourier modes and thus induce periodic eigenvectors for emergence of grid-like tuning. One reason to question this assumption is significant previous literature suggesting that place cells over-represent certain locations e.g., borders, landmarks, rewarded locations [25, 23, 33, 15, 16, 8, 6, 35, 12, 3]. Another reason is that place cells have a diversity of tuning curve widths, even at a single dorsoventral location, and even within individual cells as observed in recent experiments, e.g., [10]; this work also theoretically and computationally shows that this dual heterogenous coding scheme is more optimal in terms of encoding position than a homogenous scheme, which underlies the assumptions of the Unified Theory and makes it unlikely to hold in biology.

We push the field forward by quantitatively measuring whether place cells are translationally invariant. We use calcium imaging of place cell populations from 320 recording sessions across animals from [18] and construct spatial autocorrelation matrices $\Sigma_i \overset{\text{def}}{=} P_i P_i^T / n_p^{(i)}$, where $P_i \in \mathbb{R}^{n_x^{(i)} \times n_p^{(i)}}$ is

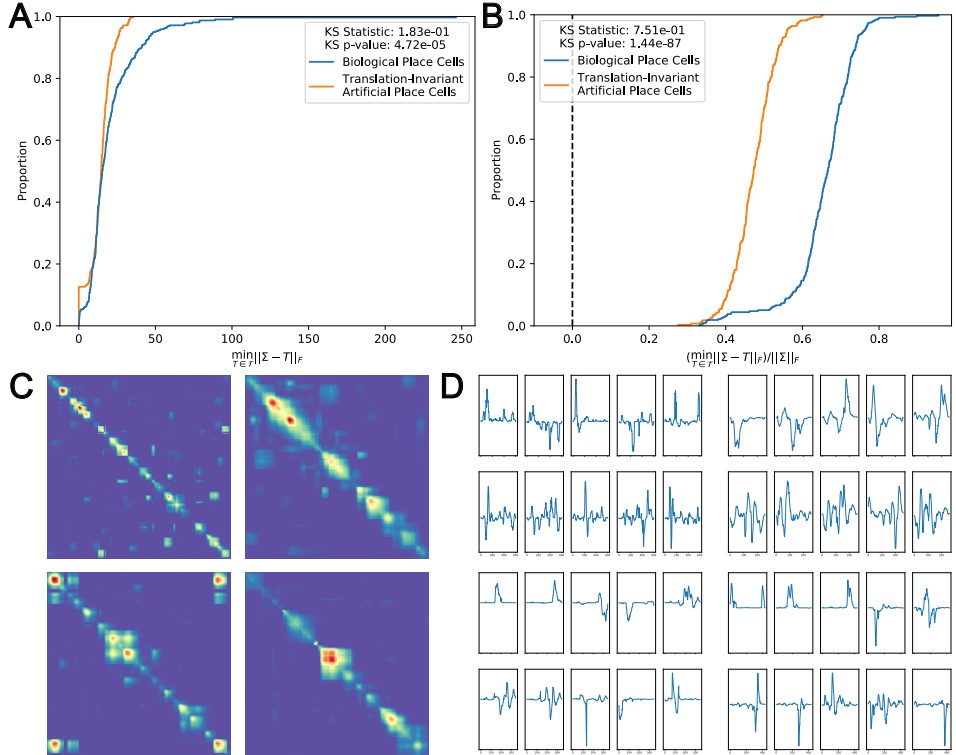

Figure 2: **Biological place cells populations are likely not translation invariant, as required mathematically by the Unified Theory.** Biological place cell populations from 320 recorded sessions [18] deviate significantly from a null distribution of translation-invariant artificial place cell population used to train deep networks, as measured by (A) the matrix distance (Kolmogorov-Smirnov 2-sample: $p = 4.72\text{e}-5$) and the (B) matrix absolute percent error (Kolmogorov-Smirnov 2-sample: $p = 1.44\text{e} - 87$). (C) Place cell spatial autocorrelation matrices $\Sigma_i$ do not visually display constant diagonals of Toeplitz matrices, as shown in 4 randomly chosen sessions. (D) Corresponding sessions' spatial autocorrelation matrices have non-periodic leading eigenvectors.

the $i$th session's $n_p^{(i)}$ place cells' signals at $n_x^{(i)}$ spatial positions. To quantify how close a spatial autocorrelation matrix is to being Toeplitz, we define a matrix's projection onto the set of Toeplitz matrices $\mathcal{T}$:

$$\Pi_{\mathcal{T}}(\Sigma_i) \overset{\text{def}}{=} \arg\min_{T \in \mathcal{T}} \left|\left|T - \Sigma_i\right|\right|_F^2. \tag{1}$$

For details, see App. C. In each of the 320 sessions, we subsample the largest continuous spatial region over which the population's summed activity is above some threshold $> 0$, construct $\Sigma_i$, then measure two different quantities to capture the extent to which the autocorrelation matrices deviates from being Toeplitz: (i) the matrix distance $\left|\left|\Pi_{\mathcal{T}}(\Sigma_i) - \Sigma_i\right|\right|_F$ and (ii) the matrix absolute percent error $\left|\left|\Pi_{\mathcal{T}}(\Sigma_i) - \Sigma_i\right|\right|_F / \left|\left|\Sigma_i\right|\right|_F$. Because our goal is to test whether biological place cells match the artificial "place cells" used to the train the networks, we constructed a null distribution based on the artificial "place cells" used to in previous papers [28, 30, 21, 31]. Specifically, we created a translation-invariant artificial place cell population comprised of 15000 single-field, single-scale, ideal-width DoG readouts, then, for each session, we randomly subsampled artificial place cells' activity $\hat{P}_i$ matching the dimensions (i.e. number of spatial bins, number of neurons) of the biological place cell activity $P_i$, computed the artificial spatial autocorrelation $\hat{\Sigma}_i \overset{\text{def}}{=} \hat{P}_i \hat{P}_i^T / n_p^{(i)}$, and measured the same two error metrics for $\hat{\Sigma}_i$. After this has been done for all 320 sessions, we apply a 2-sample Kolmogorov-Smirnov test [19] to both metrics under the null hypotheses that the biological and artificial empirical distributions were drawn from the same distribution.

For both metrics, and for all tested thresholds of spatial region coverage, biological responses have correlation structures that deviate significantly from the requisite Toeplitz structure (Fig. 2AB).

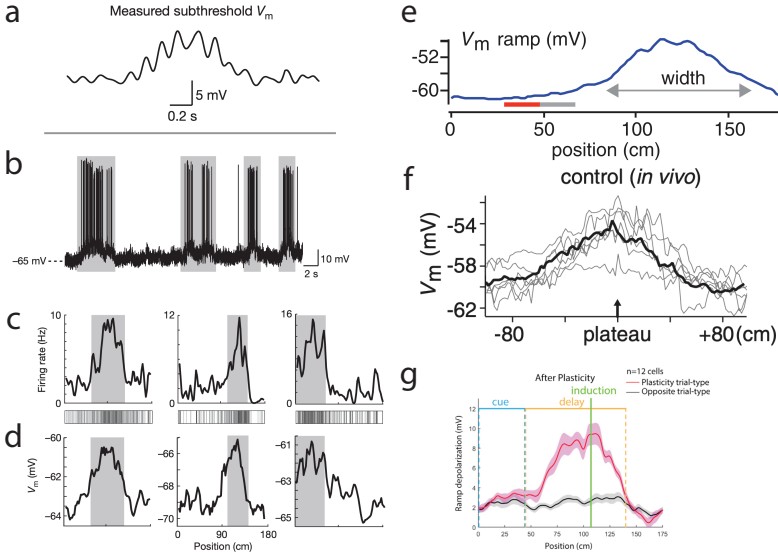

Figure 3: **Subthreshold voltages of place cells do not display DoG/DoS tuning.** (a-g) Measured membrane voltage from a selection of place cells in various conditions. Panels a-d reproduced from [14], e-f reproduced from [2] and g reproduced from [36] with permission.

Specifically, we find the probability that biological place cell spatial autocorrelation matrices comes from the same distribution as the artificial translation-invariant autocorrelation matrices is, per the matrix distance, $4.72e - 5$ (Fig. 2A) and, per the matrix absolute percent error, $1.44e - 87$ (Fig. 2B). As further confirmation, the spatial autocorrelation matrices do not visually display the constant diagonals of Toeplitz matrices (Fig. 2C), and the leading eigenvectors of the biological spatial autocorrelation matrices are not periodic (Fig. 2D). These results suggest that biological place cell populations likely lack the translation invariance required by the Unified Theory.

**Do place cells or their subthreshold responses have DoS or particular DoG shapes?** In order to explain the origin of grid cells, the Unified Theory requires that the readouts of the grid cells must individually exhibit DoS or a particular DoG tuning curve shapes (App. B). However, when testing this on biological data, there is some ambiguity: what are the readouts of biological grid cells? Earlier papers [1, 28, 30, 21, 31] referred to the readouts as place cells, e.g., Section 2 of [28] and Figure 1 of [31]. However, biological place cells do not possess DoS/DoG-shaped tuning curves, as can been seen from the wealth of extracellular place cell electrophysiology results [7, 24, 22, 34, 11, 9, 20].

A second possibility later suggested by the Unified Theory is that the readouts are *subthreshold inputs to place cells from grid cells* [31, 29], though it is difficult to imagine why or how the target for the entorhinal grid cells would be a particular shape of *subthreshold* activation function of downstream neurons. To test this possibility, we hunted down and collated intracellular voltage recordings of CA1 place cells [14, 2, 36] (Fig. 3). These recordings do not reveal DoS/DoG-shaped subthreshold responses near their place fields. These results suggest that place cell inputs likely lack the center-surround shape required by the Unified Theory and used numerically [28, 30, 21, 31]. A third possibility suggested by the Unified Theory is that readouts are summed grid cell contribution to inputs to place cells; we do not know of any dataset or experimental technique through which this claim could be tested.

## 3   Discussion

In this work, we test two assumptions made by the Unified Theory for the origin of grid cells and found both are likely not true at both the population and single cell levels. Our results call the Unified Theory into question, suggesting that biological grid cells likely have a different origin than the

grid-like representations found in trained artificial neural networks [1, 5, 28]. For a more promising alternative, see [27].

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

## A  Tuning Curves: Difference of Softmaxes and Difference of Gaussians

What is a Difference-of-Softmaxes (DoS) or Difference-of-Gaussians (DoG) tuning curve? Suppose we sample a sequence of positions $x_0, ..., x_T \in \mathbb{R}^2$. We sample $N_p$ place cell centers $\{p_i\}_{i=1}^{N_p}$ uniformly at random within the (bounded) environment.

- **Difference of Gaussians:** A vector in $\mathbb{R}^{N_p}$ whose entries are given by:

$$\alpha_E \exp\Big( -\frac{1}{2\sigma_E^2} ||x_t - p_i||^2 \Big) - \alpha_I \exp\Big( -\frac{1}{2\sigma_I^2} ||x_t - p_i||^2 \Big)$$

- **Difference of Softmaxes:** A vector in $\mathbb{R}^{N_p}$ whose entries are given by:

$$Softmax\Big( -\frac{1}{2\sigma_E^2} ||x_t - p_i||^2 \Big) - Softmax\Big( -\frac{1}{2\sigma_I^2} ||x_t - p_i||^2 \Big)$$

In the Unified Theory, for DoG, only certain combinations of $(\alpha_E, \sigma_E, \alpha_I, \sigma_I)$ should be theoretically expected to produce grid-like tuning; most will not. See Fig. 4C in [26] for more details.

## B  Reproduction of & Commentary on the Unified Theory for the Origin of Grid Cells

Here, we reproduce the Unified Theory of [28, 30, 31] to elucidate its assumptions, including the two assumptions that we test in this work: (1) place cells as a population are translationally invariant, and (2) place cells (or their subthreshold inputs) have difference-of-Gaussian or difference-of-Softmaxes center-surround tuning curves. We note that the Unified Theory does not deal with dynamics of path integration or learning dynamics of a deep recurrent network, but rather concerns the problem of readout reconstruction/prediction. This leads us to the first assumption:

**Assumption 1 (A1):** The hypothetical network representations $G \in \mathbb{R}^{n_x \times n_g}$ is some function of space. Here $n_x$ is the number of spatial locations and $n_g$ is the number of hidden units. This is a subtle but significant assumption because, for recurrent networks given velocity inputs, the networks' representations are not a function of space, but rather develop into a function of space (i.e. builds a continuous attractor) over the course of training. For a better understanding of why the assumption of building a continuous manifold of fixed points is significant, see literature of the theory of continuous attractors which is briefly reviewed in [17].

Under A1, consider a feedforward mapping $\hat{P} \overset{\text{def}}{=} GW$ where $W \in \mathbb{R}^{n_g \times n_p}$. Here $n_p$ is the number of readout units. One can define the readout reconstruction error as the mean square loss between the readout target $P \in \mathbb{R}^{n_x \times n_p}$ and prediction $\hat{P} \overset{\text{def}}{=} GW$ :

$$\mathcal{E}(G, W) \overset{\text{def}}{=} ||P - \hat{P}||_F^2 = ||P - GW||_F^2 \tag{2}$$

**Assumption 2 (A2):** Linear readout $W$ relaxes, reaching its optimum much faster than $G$ changes, so that we can replace $W$ with its optimal ordinary least squares value for fixed $G, P$:

$$W^*(G, P) = (G^T G)^{-1} G^T P \tag{3}$$

Substituting $W^*(G, P)$ into the loss for $W$ yields the error as a function of $P$ and $G$:

$$\mathcal{E}(G, P) = ||P - G(G^T G)^{-1} G^T P||_F^2 \tag{4}$$

**Assumption 3 (A3):** $G$'s columns can be made orthonormal i.e. $G^T G = I_{n_g}$. This means that each grid unit has the same average firing rate over all space, and that any 2 grid cells do not overlap.

Then, we can write down a Lagrangian for this optimization problem with Lagrange multiplier $\lambda$ for this constraint:

$$\mathcal{L} = -\mathcal{E}(G, P) - \lambda(G^T G - I_{n_g}) \tag{5}$$

We can then set $G^T G$ to $I$ in the error term and the Lagrangian is now written as:

$$\mathcal{L} = -||P - GG^T P||_F^2 - \lambda(G^T G - I_{n_g}) \tag{6}$$

$$\mathcal{L} = -\text{Tr}[(P - GG^T P)^T(P - GG^T P)] - \lambda(G^T G - I_{n_g}) \tag{7}$$

Here, the identity $||M||_F^2 = \text{Tr}(M^T M)$ has been employed. The trace term can be simplified further using the cyclic permutation property of Trace: $\text{Tr}(ABC) = \text{Tr}(CAB)$,

$$\text{Tr}[(P - GG^T P)^T(P - GG^T P)] = \text{Tr}(P^T P) + \text{Tr}[G^T(PP^T)G(G^T G)] - 2\text{Tr}(G^T PP^T G)$$

Here $\Sigma \overset{\text{def}}{=} \frac{1}{n_p} PP^T \in \mathbb{R}^{n_x \times n_x}$ is the readout spatial correlation matrix. We can also drop the $G$ independent term above. Using the trace identity again, this term simplifies to $\text{Tr}(G^T PP^T G)$. Hence the total simplified Lagrangian is then:

$$\mathcal{L} = \text{Tr}[G^T \Sigma G - \lambda(G^T G - I_{n_g})] \tag{8}$$

Considering gradient learning dynamics, one gets the following evolution equation for $G$:

$$\frac{d}{dt}G = \nabla_G \mathcal{L} \Rightarrow \frac{d}{dt}G = \Sigma G - \lambda G \tag{9}$$

[28, 30] then simplify further analysis by considering a single grid unit. This corresponds to replacing the $n_x \times n_g$ matrix $G$ by the $n_x \times 1$ column vector $g$:

$$\frac{d}{dt}g = \Sigma g - \lambda g \tag{10}$$

This linear dynamical system captures how the pattern $g$ of a unit evolves with gradient learning. The Unified Theory concludes that the eigenvectors corresponding to the subspace of the *top eigenvalue* form the optimal pattern, since these eigenvectors will grow exponentially with the fastest rate.

**Assumption 4 (A4)**: The readout spatial correlation $\Sigma$ is translation-invariant over space i.e. $\Sigma_{x,x'} = \frac{1}{n_p}\sum_{i=1}^{n_p} p_i(x)p_i(x') = \frac{1}{n_p}\sum_{i=1}^{n_p} p_i(x + \Delta)p_i(x' + \Delta) = \Sigma_{x+\Delta,x'+\Delta} \forall \Delta$.

**Assumption 5 (A5)**: The environment has periodic boundaries (or no boundaries, which corresponds to a continuum limit). [An alternative assumption in other parts of the derivations and numerics is **(A5')**: the assumption involves periodic boundary conditions with a small box size $L$.]

Under A4 and A5, the eigenmodes of $\Sigma$ are exactly Fourier modes across space and form a periodic basis. The normalized eigenvectors are indexed by their wavelength, $\mathbf{k}$, and are denoted $f_\mathbf{k}$ with corresponding eigenvalue $\lambda_\mathbf{k}$. To calculate this eigenvalue, the Unified Theory uses Fourier analysis:

$$\Sigma f_\mathbf{k} = \lambda_\mathbf{k} f_\mathbf{k}$$
$$\implies \lambda_\mathbf{k} = f_\mathbf{k}^\dagger \Sigma f_\mathbf{k}$$

Here $f_\mathbf{k}^\dagger$ denotes the conjugate of the eigenvector $f_\mathbf{k}$.

Next, they rewrite in component form: $\Sigma_{x,x'} = 1/n_p(PP^T)_{x,x'} = 1/n_p \sum_{i=1}^{n_p} p_i(x)p_i(x')$

$$\lambda_\mathbf{k} = f_\mathbf{k}^\dagger \Sigma f_\mathbf{k} = \sum_{i=1}^{n_p} \sum_{x,x'} \frac{1}{n_p} f_\mathbf{k}^*(x')p_i(x)p_i(x')f_\mathbf{k}(x)$$

$$= \frac{1}{n_p}\sum_{i=1}^{n_p}\left(\sum_x p_i(x)f_\mathbf{k}^*(x)\right)\left(\sum_{x'} p_i(x')f_\mathbf{k}(x')\right)$$

$$= \frac{1}{n_p}\sum_{i=1}^{n_p} \tilde{p}^*(k)\tilde{p}(k) = |\tilde{p}(k)|^2$$

$$\implies \lambda_\mathbf{k} = |\tilde{p}(k)|^2$$

The Unified Theory concludes that the eigenvalue corresponding to eigenvectors with wavelength $k$ is given by the corresponding power of the Fourier spectrum of the readout correlation matrix $\Sigma$. The optimal pattern is thus the one which has the highest Fourier power in $\Sigma$.

Further, [28, 30] consider the effect of non-negativity perturbatively in the readout regression framework by phenomenologically adding a term to the Lagrangian, $= \int_x \sigma(g)dx$.

$$\mathcal{L} = \text{Tr}\big[G^T\Sigma G - \lambda(G^T G - I_{n_g})\big] + \int_x \sigma(g)dx \tag{11}$$

In Fourier space, [28, 30] show perturbatively that this amounts to a cubic interaction term, which is the leading order term that non-trivially distinguishes between nonlinearities such as ReLU and Sigmoid which break the $g \mapsto -g$ symmmetry and nonlinearities such as Tanh which do not. Again, specializing to the single neuron Lagrangian,

$$\mathcal{L}_{int} = \int_{k,k',k''} g(k)g(k')g(k'')\delta(k + k' + k'')dkdk'dk''$$

This term effectively acts as a penalty for non-negativity. Here, it is important to point out that this cubic term appears not only for non-negativity, but rather *any* function that is not anti-symmetric. Negative activation functions such as slightly shifted Tanh can also have a cubic term. Thus, non-negativity is a special case, as has been noted in [28] Appendix B. We refer to this as **Assumption 6 (A6)**.

Under this assumption, [28, 30] conclude that the optimal pattern consists of a triplet of Fourier waves with equal amplitude and $k-$vectors that lie on an equilateral triangle, at $60^o$ to each other.

Next, we examine the Fourier spectrum of a translationally invariant Gaussian readout $f(\Delta x) = \frac{1}{\sqrt{2\pi\sigma^2}}\exp(-(\Delta x^2)/2\sigma^2)$ under the assumptions of this theory. For simplicity and to provide intuition we write its Fourier transform in 1d, which is given by another Gaussian. The peak of the Fourier spectrum is at $k = 0$, or the DC, non-periodic mode:

$$\tilde{f}(k) = \int_{\mathcal{R}} \frac{1}{\sqrt{2\pi\sigma^2}}\exp(-(\Delta x^2)/2\sigma^2)e^{ik\Delta x}d\Delta x$$
$$= \exp(-k^2\sigma^2/2)$$

In simulations in a finite environment of length $L$, the allowed frequency modes are discretized with bin-size $2\pi/L$ (**(A5')**). This is shown in Fig.4 as lattice points with the Fourier spectrum overlayed as in [28, 30]. Gaussian readouts produce a Fourier spectrum peaked at the central DC mode. This mode has no periodicity and thus the theory for a single hidden unit predicts no lattices in this hidden unit, in the continuum or small-box discrete limit.

Until now, all analysis was performed for a single grid unit. What happens in full multi-cell setting? For this case, [28] shows that the global optimum to the constrained optimization problem:

$$\max_G \text{Tr}(G^T\Sigma G) \text{ such that } G^T G = I_{n_g},$$

i.e. under **(A3)** involves the columns of $G$ spanning the top $n_g$ eigenmodes of $\Sigma$ (Theorem B.2, Appendix B of [28]). Under **(A6)**, [28] also shows that with a Fourier spectrum consisting of a wide annulus (in the discrete setting this corresponds to a Fourier spectrum of rings of different radii), the optimum consists of a hierarchy of hexagonal maps, but only if the Fourier powers of these rings are exactly equal (Lemma B.3, Appendix B). For purely Gaussian spectra (corresponding to Gaussian readouts), the full theoretical solution of this problem depends on specific details of the power spectrum curve (i.e. the width of the Gaussian) since each discrete eigenmode has a different Fourier power which must be taken into account while constructing linear combinations of modes, and thus is not solvable analytically. Instead, [28] numerically simulates the Lagrangian dynamics with assumption **(A3)** to find a hierarchy of lattices. However, the number of different period lattices that result is related directly to and nearly as numerous as the number of cells. For a large number of neurons, such as the 4096 hidden units used in simulations [31, 29], and sufficient wide power spectra, this would mean the optimum solutions would consist of likely hundreds of discrete frequencies (each corresponding to a grid module). This is to be contrasted with the 4-8 modules estimated to exist in rodents [32].

We next consider a translationally invariant Difference-of-Gaussians readout. We refer to this as **Assumption 7 (A7)**.

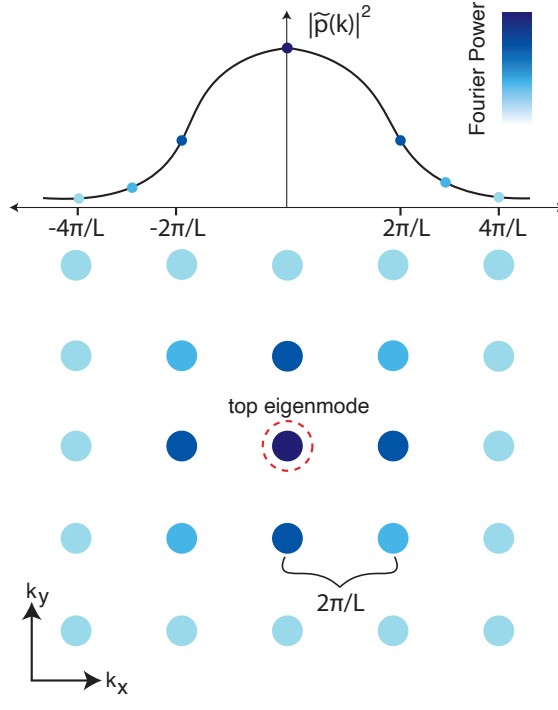

Figure 4: Fourier structure of Gaussian readouts.

$$f(\Delta x) = \frac{\alpha_E}{\sqrt{2\pi\sigma_E^2}} \exp(-(\Delta x)^2/2\sigma_E^2) - \frac{\alpha_I}{\sqrt{2\pi\sigma_I^2}} \exp(-(\Delta x)^2/2\sigma_I^2)$$

Under A7, the readout Fourier spectrum is given by:

$$\begin{aligned}
\tilde{f}(k) &= \int_{\mathcal{R}} d(\Delta x) f(\Delta x) e^{ik\Delta x} \\
&= \alpha_E \sigma_E \exp(-\sigma_E^2 k^2/2) - \alpha_I \sigma_I \exp(-\sigma_I^2 k^2/2)
\end{aligned}$$

The solution will be periodic if the maximum, given by $[k^*]^2 = \frac{2}{\sigma_E^2 - \sigma_I^2} \log(\alpha_E \sigma_E^3/\alpha_I \sigma_I^3)$, contains sufficient power and if $k^* \neq 0$; specifically, the condition for pattern formation is $\tilde{f}(k) > 1$; see [4, 17, 26] for more details. This reveals the second assumption that we focus on in this work: the Unified Theory requires that readout tuning curves have a center-surround functional form with hyperparameters $\alpha_E, \alpha_I, \sigma_E, \sigma_I$ lying in a narrow range.

## C   Projecting a Matrix onto Set of Toeplitz Matrices

In the main text, we needed to quantify how close a spatial autocorrelation matrix is to being Toeplitz. We thus define a matrix's projection onto the set of Toeplitz matrices $\mathcal{T}$ in Eqn. 1, repeated here for convenience:

$$\Pi_{\mathcal{T}}(\Sigma_i) \stackrel{\text{def}}{=} \arg\min_{T \in \mathcal{T}} \left\| T - \Sigma_i \right\|_F^2. \tag{12}$$

How is this projection performed? Recall that a Toeplitz matrix is defined as a diagonal-constant matrix, i.e., each diagonal has a constant value. Consequently, to project $\Sigma_i$ onto the set of Topelitz matrices, for each diagonal, we compute $\Sigma_i$'s average value and take that average value to be the

Toeplitz matrix's constant value. For a small example:

$$\Pi_{\mathcal{T}}\left(\begin{bmatrix} 1 & 4 & 7 \\ 2 & 5 & 8 \\ 3 & 6 & 9 \end{bmatrix}\right) = \begin{bmatrix} (1+5+9)/3 & (4+8)/2 & (7)/1 \\ (2+6)/2 & (1+5+9)/3 & (4+8)/2 \\ (3)/1 & (2+6)/2 & (1+5+9)/3 \end{bmatrix}$$

