# OpenReview forum: "Testing Assumptions Underlying a Unified Theory for the Origin of Grid Cells"
_NeurIPS.cc/2023/Workshop/AI4Science — NeurIPS2023-AI4Science Poster_

### Official Review · Reviewer_yHNm · 2023-10-23
**A limited comparison of artificial and biological place cells**

**Rating:** 6
**Confidence:** 4

**Review:**

This study tested the assumption underlying the “Unified Theory” for the origin of grid cells using the biological place cells generated from previous in vivo two-photon calcium imaging experiments. The authors claimed that as the primary/putative readouts of grid cells, the “unified theory” assumed that place cells should be translationally invariant as a population and center-surround tuned as an individual. The authors found that the biological place cells do not have translational invariant properties (Figure 2) nor center-surround inputs (Figure 3).

Pros: It’s very interesting and important to compare the artificial place cells with the biological place cells. I am not very confident about the mathematical assumption of “unified theory”. If it is true, I believe the data from the two figures basically confirmed that the artificial place cells’ properties are not consistent with real biological place cells.

Cons: The data analysis in the two figures is not very thorough enough, especially for Figure 3 which is more like a literature review for Introduction or Discussion instead of real Results. It would be great if the authors could get the experimental data from the authors who generated those data.

Furthermore, there are some concerns about using virtual reality to study the grid and place cells (Minderer, Harvey, Donato & Moser, Nature 2016). Biological place cells in this study are generated from head-fixed mice, which are kinds of “artificial” when compared to the place cells generated from free-moving rodents.

Line 57 “One reason to suspect this assumption is incorrect is significant previous literature suggesting place…” is hard to understand.

---

### Official Review · Reviewer_tUSK · 2023-10-24
**Challenging Assumptions: A Critical Examination of the Unified Theory for the Origin of Grid Cells**

**Rating:** 8
**Confidence:** 3

**Review:**

The authors critically examine two fundamental assumptions of the Unified Theory for the Origin of Grid Cells and have designed experiments to validate them. Intriguingly, their findings challenge the Unified Theory, with the authors positing that biological grid cells may have a distinct origin from the grid-like representations observed in trained artificial neural networks.

The paper is coherently structured with robust experimentation and analysis. While the subject matter is outside my primary areas of expertise (Genomics and ML), I would recommend acceptance of this paper, unless other reviewers identify factual inaccuracies or controversies.

---

### Meta-Review · Program_Chairs · 2023-10-27

**Recommendation:** Accept (Poster)
**Confidence:** 3

**Metareview:**

SUMMARY: Both reviewers converge towards accepting the paper and agree that the subject content (testing the mathematical assumptions of posited Unified Theories of Grid Cell behavior experimentally) is very interesting and relevant to the workshop. Reviewers expressed mixed opinions on the thoroughness of the experiments. Upon reading the manuscript and reviews, I lean towards acceptance of the paper and look forward to the followup discussion it will generate at the workshop if accepted.

SUGGESTIONS/COMMENTS:

1. Please clarify Reviewer yHNMs concerns about the experimental conditions and data collection in the manuscript.
2. Upon reading Section 2.2, it is not immediately obvious how the projection step to Toeplitz matrix manifolds is performed in Eq. (1). It would also be good to provide a few lines of justification on why the l2 distance (Eq. 1) is the metric of choice for projection as well as for evaluating deviation from the assumption, especially since autocorrelation matrices have a very specific PSD structure by definition.
3. Please clarify what the authors mean by "over-represent" in line 57.